

# CRISPR 2 PCR and high resolution melting profiling for identification and characterization of clinically-relevant *Salmonella* enterica subsp. enterica

Nuttachat Wisittipanit[1,*], Chaiwat Pulsrikarn[2], Sudarat Srisong[3], Rungthiwa Srimora[3], Nattinee Kittiwan[4] and Kritchai Poonchareon[3,*]

[1] Department of Material Engineering, School of Science, Mae Fah Luang University, Chiang Rai, Thailand
[2] Department of Medical Sciences, WHO National Salmonella and Shigella Center, National Institute of Health, Ministry of Public Health, Nonthaburi, Thailand
[3] Division of Biochemistry, School of Medical Sciences, University of Phayao, Phayao, Thailand
[4] Veterinary Research and Development Center (Upper Northern Region), Lampang, Thailand
[*] These authors contributed equally to this work.

Corresponding author
Kritchai Poonchareon,
kof_of@hotmail.com

## ABSTRACT

**Background**. Nontyphoidal *Salmonella* spp. constitute a major bacterial cause of food poisoning. Each *Salmonella* serotype causes distinct virulence to humans.

**Method**. A small cohort study was conducted to characterize several aspects of *Salmonella* isolates obtained from stool of diarrheal patients ($n = 26$) admitted to Phayao Ram Hospital, Phayao province, Thailand. A simple CRISPR 2 molecular analysis was developed to rapidly type *Salmonella* isolates employing both uniplex and high resolution melting (HRM) curve analysis.

**Results**. CRISPR 2 monoplex PCR generated a single *Salmonella* serotype-specific amplicon, showing *S.* 4,[5],12:i:- with highest frequency (42%), *S.* Enteritidis (15%) and *S.* Stanley (11%); *S.* Typhimurium was not detected. CRISPR 2 HRM-PCR allowed further classification of *S.* 4,[5],12:i:- isolates based on their specific CRISPR 2 signature sequences. The highest prevalence of *Salmonella* infection was during the summer season (April to August). Additional studies were conducted using standard multiplex HRM-PCR typing, which confirmed CRISPR 2 PCR results and, using a machine-learning algorithm, clustered the majority of *Salmonella* serotypes into six clades; repetitive element-based (ERIC) PCR, which clustered the serotypes into three clades only; antibiogram profiling, which revealed the majority resistant to ampicillin (69%); and test for extended spectrum $\beta$-lactamase production (two isolates) and PCR-based detection of *bla* alleles.

**Conclusion**. CRISPR 2 PCR provided a simple assay for detection and identification of clinically-relevant *Salmonella* serotypes. In conjunction with antibiogram profiling and rapid assay for $\beta$-lactamase producers, this approach should facilitate detection and appropriate treatment of Salmonellosis in a local hospital setting. In addition, CRISPR 2 HRM-PCR profiling enabled clustering of *S.* 4,[5],12:i:-isolates according to CRISPR 2 locus signature sequences, indicative of their different evolutionary trajectories, thereby providing a powerful tool for future epidemiological studies of virulent *Salmonella* serotypes.

## INTRODUCTION

*Salmonella* enterica can cause human gastroenteritis owing to inadequate hygienic standards of living and/or consuming poorly prepared fresh food. *S.* enterica has been subtyped to more than 2,463 serovars according to their antigenic properties (*Kauffman, 1972*); however, only a limited number of serovars cause human infection (*Uzzau et al., 2000*) and, thus, identification of these serovars is of public health importance. Distribution of *Salmonella* serovars is markedly different among continents (*Galanis et al., 2006*), but within a single country such as Thailand, *Salmonella* serovars slightly varies in different geographical regions, the most common clinical serovars being *S.* 4,[5],12:i:-, *S.* Anatum, *S.* Derby, *S.* Enteritidis, and *S.* Weltevreden (*Bangtrakulnonth et al., 2004*; *Padungtod & Kaneene, 2006*).

Emergence of multi-drug resistance (MDR) in the major prevalent non-typhoidal *Salmonella* serovars found in human gastroenteritis is related to an acquisition of antibiotic-resistant genes especially in *S.* Typhimurium (*Velge, Cloeckart & Barrow, 2005*). In Thailand, a significant increase in resistance to amikacin, kanamycin and second line antibiotic ceftriaxone is suggested to be related to overuse of antibiotics to livestock feed and in medical treatment (*Boonmar et al., 1998*). Also, an increase in resistance to quinolones such as nalidixic acid was reported (*Sirsichote et al., 2010*). Failure of treatment from third-generation cephalosporins, the extended spectrum $\beta$-lactamases (ESBLs), has created a major threat to treatment of MDR microbial pathogens and has stimulated studies on their epidemiology (*Rupp & Fey, 2003*). ESBL genes, such as *bla*OXA, *bla*SHV, *bla*TEM and the more recent *bla*CTX-M, generate different levels of virulence and transmission capability (*Ewers et al., 2012*). ESBL $\beta$-lactamase CTX-M is of particular concern due to its global dissemination through both clonal and horizontal gene transfer (*Cantón, González-Alba & Galán, 2012*).

The standard method for *Salmonella* identification is based on a culture method (*ISO, 2002*). The assay uses different selective media, followed by serotyping based on various combinations of O and H antigenic determinants (*Kauffman, 1972*). These methods require skilled personnel to perform the standardized protocols, which are laborious and time-consuming. In order to reduce the turnover time for typing *Salmonella* spp., multiplex PCR of specific gene determinants was introduced, which depends on sequence polymorphisms of *rfb* locus and flagellar alleles (*Kim et al., 2006*; *Masek et al., 2014*). Other molecular modifications such as high resolution melting temperature (HRM)-PCR assay has been coupled to multiplex PCR for detection of polymorphisms of 16S rDNA (*O'Regan et al., 2008*), *flj*B, *gyr*B and *ycf*Q (*Zeinzinger et al., 2012*). HRM-PCR was applied to discriminate between two most prevalent clinical serotypes, namely, *S.* Enteritidis and *S.* Typhimurium (*Bratčikov & Mauricas, 2009*).

Clustered regularly interspaced short palindromic repeats (CRISPR) were discovered as a new family of repeated DNA sequences in many prokaryotes (*Jansen et al., 2002*). Their genetic signatures are characterized by repeated patterns of DNA, known as direct repeats (DRs), 24–47 bp in length, and DNA variable sequences (spacers) of 21–72 bp (*Shariat & Dudley, 2014*). Adjacent to the CRISPR locus are a "leader sequence" and cas (CRISPR-associated sequence) gene (*Horvath & Barrangou, 2010*). *Salmonella* spp. harbor two different CRISPR loci, namely, CRISPR 1 and CRISPR 2 that correlate with *Salmonella* serotype and multilocus sequence type (*Fabre et al., 2012*). Characterization of CRISPR alleles provide information of spacer content for performing *Salmonella* typing and subtyping (*Fabre et al., 2012*; *Liu et al., 2011*). CRISPR typing was recently applied to identify virulent *Salmonella* infection of chicken raised in a single farm as well as in farms from different areas (*Fei et al., 2017*; *Li et al., 2018*). More recently, CRISPR loci polymorphisms were utilized in a single-step assay to identify multiple *Salmonella* spp. contamination of a poultry sample (*Thompson et al., 2018*).

In order to reduce operational cost and turnover time of the traditional culture assay for detecting and typing *Salmonella* spp., a simple CRISPR 2-based conventional and monoplex HRM-PCR was developed in conjunction with HRM triplex and a novel machine-learning algorithm tool to analyze and identify *Salmonella* serotypes in correlation with repetitive element-based (ERIC)-PCR genotyping and a rapid ESBL test, and, in addition, to distinguish *S.* 4,[5],12:i:- isolates from different epidemiological settings.

## MATERIALS & METHODS

### Isolation and identification of *Salmonella* isolates in stool of diarrheal patients

Stool samples were collected from patients ($n = 26$) with diarrhea during May 2016 to July 2017 at Phayao Ram Hospital, Phayao province, Thailand. Diarrhea is defined as patients showing related symptoms of fecal incontinence diagnosed by a physician and participants signed the consent forms after brief explanation of the research project. Samples were incubated in buffered peptone water (Oxoid, Hampshire, UK) at 37 °C overnight, plated on SS and XLD agar (Oxoid, Hampshire, UK), and suspected *Salmonella* colonies indicated by using a triple sugar iron (TSI) slant (Oxoid, Hampshire, UK) assay and lysine iron agar (LIA) (Biomedia, Nontanuri, Thailand). Conventional biochemical tests (glucose fermentation, lactose oxidation, gas and $H_2S$ production, urease assay, methyl red staining, and indole, motility and Voges-Proskauer tests) were performed according to ISO 6579:2002.

The study protocols were approved by the Ethical Committee of Phayao University (no. 57 02 04 0020).

### Determination of $\beta$-lactam antibiotic resistance profile and ESBL phenotype

Susceptibility to $\beta$-lactam antibiotics was performed using a disk diffusion method according to Clinical and Laboratory Standards Institute (CLSI) guidelines (*CLSI, 2016*) using ampicillin (AMP, l0 µg), cefotaxime (CTX, 30 µg), ciprofloxacin (CIP, 5 µg),

ertapenem (ETP, 10 µg), and gentamycin (GEN, 10 µg) discs (Oxoid, Hampshire, UK), with *Escherichia coli* ATCC 25922 as control. ESBL phenotype was evaluated using a double-dish method (*CLSI, 2016*) employing CTX alone and in combination with clavulanic acid (10 µg) (Oxoid, Hampshire, UK), with in-house known ESBL-producing and -negative *Escherichia coli* strains as controls. A rapid ESBL was slightly modified from the ESBL NDP (*Nordmann, Dortet & Poirel, 2012*), which is based on observation of color change due to acid production of ESBL-producing bacteria. *Salmonella* isolate was inoculated in one ml of NB broth (Oxoid, Hampshire, UK) and incubated at 37 °C overnight, each experiment conducted in duplicate. Then culture was centrifuged at 13,000 g for 2 min, pellet washed twice with 700 µL of 1X 10 mM Tris HCl pH 8.0 containing 1 mM EDTA (TE) buffer and mixed with 100 µL of lysis buffer [(B-PERII Bacterial Protein Extraction Reagent)] (Pierce/Thermo Scientific, Villebon-sur-Yvette, France), followed by 100 µL of revelation solution (0.05% (w/v) phenol red) with and without 6 g/ml cefotaxime and incubation at 37 °C for 10 min; yellow color in the former indicating positive result.

## Determination of *Salmonella* serotypes by HRM-PCR

DNA was extracted from *Salmonella* isolates as previously described (*McNerney et al., 2017*). In brief, one ml aliquot of an overnight culture was sedimented as described above, washed twice with 400 µL of TE buffer, resuspended in the same volume of TE buffer, incubated at 80 °C for 20 min, and cooled to ambient temperature. Then, a 50 µL aliquot of lysozyme solution (10 mg/mL) was added and the solution incubated at 37 °C for one hour with occasionally shaking, followed by addition of 75 µL of 10% SDS/proteinase K (10 mg/mL) solution, vigorous vertexing and incubation at 65 °C for 10 min. Following addition of 100 µL of 5 M NaCl and 100 µL of prewarmed (65 °C) 10% N-cetyl-N,N,N,-trimethyl ammonium bromide (CTAB)/ 5 M NaCl solution, the mixture was further incubated at 65 °C for 10 min, followed by addition of 750 µL of chloroform : isoamyl alcohol (24 :1) and centrifugation at 11,000 g at 4 °C for 5 min. DNA in the upper aqueous was precipitated with ethanol, resuspended in 50 µL of double-distilled water and stored at −20 °C until used.

Multiplex HRM-PCR was performed using a combination of primers to amplify *flj*B (170 bp), *gyr*B (171 bp) and *ycf*Q (241 bp) (Table 1). HRM-PCR mixture (10 µL) contained 1 µL of DNA, 0.1 pmol of gyrB, 0.075 pmol of fljB and 0.075 pmol of ycfQ primer pairs and 2 µL of HOT FIREPol EvaGreen: no ROX Mix (Solis Biodye, Tartu, Estonia). Thermocycling was performed in a BIO-RAD CFX96$^{TM}$ Real-Time System (Bio-Rad, Hercules, CA, USA) as follows: 95 °C for 12 min, followed by 40 cycles of 95 °C for 10 s, 60 °C for 10 s and 72 °C for 20 s. Samples were then heated at 95 °C for 1 min, cooled to 40 °C for 1 min and then heated from 70 to 95 °C at 0.2 /s, with 25 fluorescence data acquisitions/°C. HRM profiles were generated using a Precision Melt Analysis software V 1.2 (BIO-RAD, Hercules, CA, USA) with sensitivity setting at 0.30, temperature shift at threshold 5, pre-melt normalization range from 80.87 to 81.51 °C, and post-melt normalization range from 87.17 to 87.92 °C. Following normalizing and temperature shifting, difference plots were generated relative to HRM profile of *S.* Bareilly (as baseline).

**Table 1   Primers used in this study.**

| Primer | Genes | Sequence (5′ → 3′) | Size of PCR-product (bp) | Primer concentration (pmol/ul) | Reference |
|---|---|---|---|---|---|
| **HRM Multiplex** *fljB, gyrB* and *ycf* Q genes (HRM-rt PCR) | | | | | |
| *fljB* _f | *fljB* | GTGAAAGATACAGCAGTAACAACG | 170 | 0.075 | *Zeinzinger et al. (2012)* |
| *fljB* _r | | CAAAGTACTTGTTATTATCTGCG | | 0.075 | *Zeinzinger et al. (2012)* |
| *gyrB* _f | *gyrB* | AAACGCCGATCCACCCGA | 171 | 0.1 | *Zeinzinger et al. (2012)* |
| *gyrB* _r | | TCATCGCCGCACGGAAG | | 0.1 | *Zeinzinger et al. (2012)* |
| *ycfQ* _f | *ycfQ* | GCCTACTCTCTATGCGGAATTCAC | 241 | 0.075 | *Zeinzinger et al. (2012)* |
| *ycfQ* _r | | GATATCGCGCGAGGAGGCG | | 0.075 | *Zeinzinger et al. (2012)* |
| **CRISPR 2 Monoplex** | | | | | |
| B1 | CRISPR 2 loci | GAGCAATACYYTRATCGTTAACGCC | Variable | 0.2 | *Fabre et al. (2012)* |
| B2 | | GTTGCDATAKGTYGRTRGRATGTRG | | 0.2 | *Fabre et al. (2012)* |
| **Multiplex 1** *bla*TEM variants including TEM-1 and TEM-2, *bla*SHV variants including SHV-1, *bla*OXA-1-like including OXA-1, OXA-4 and OXA-30 | | | | | |
| TEM_f | *bla*TEM | CATTTCCGTGTCGCCCTTATTC | 800 | 0.4 | *Dallenne et al. (2010)* |
| TEM_r | | CGTTCATCCATAGTTGCCTGAC | | 0.4 | *Dallenne et al. (2010)* |
| SHV_f | *bla*SHV | AGCCGCTTGAGCAAATTAAAC | 713 | 0.4 | *Dallenne et al. (2010)* |
| SHV_r | | ATCCCGCAGATAAATCACCAC | | 0.4 | *Dallenne et al. (2010)* |
| OXA_f | *bla*OXA | GGCACCAGATTCAACTTTCAAG | 564 | 0.4 | *Dallenne et al. (2010)* |
| OXA_r | | GACCCCAAGTTTCCTGTAAGTG | | 0.4 | *Dallenne et al. (2010)* |
| **Multiplex 2** *bla*CTX-M group 1 and group 9; variants of *bla*CTX-M group 1 including CTX-M-1, CTX-M-3 and CTX-M-15, variants of *bla*CTX-M group 9 including CTX-M-9 and CTX-M-14 | | | | | |
| CTX 1_f | *bla*CTX-M group 1 | TTAGGAARTGTGCCGCTGYA[a] | 688 | 0.4 | *Dallenne et al. (2010)* |
| CTX 1_r | | CGATATCGTTGGTGGTRCCAT[a] | | 0.2 | *Dallenne et al. (2010)* |
| CTX 9_f | *bla*CTX-M group 9 | TCAAGCCTGCCGATCTGGT | 561 | 0.4 | *Dallenne et al. (2010)* |
| CTX 9_r | | TGATTCTCGCCGCTGAAG | | 0.4 | *Dallenne et al. (2010)* |
| **Molecular typing** | | | | | |
| ERIC_f | ERIC-PCR | ATGTAAGCTCCTGGGGATTCAC | Variable | 20 | *Versalovic, Koeuth & Lupski (1991)* |
| ERIC_r | | AAGTAAGTGACTGGGGTGAGCG | | 20 | *Versalovic, Koeuth & Lupski (1991)* |

Notes.
[a] Y = T or C; R = A or G; S = G or C; D = A or G or T.

## Hierarchical clustering of HRM curves using dynamic time warping (DTW) algorithm

A dendrogram of normalized HRM curves was constructed using a DTW algorithm to determine distance measurements (*Lu et al., 2017*); the entire dendrogram construction was performed in Python programming language (*Van Rossum, 1995*). In short, a smooth spline approximation was determined from each normalized HRM curve using cubic splines of splrep function in scipy module (*Jones, Oliphant & Peterson, 2001*) followed by a rate curve calculated from negative first derivative of the resulting spline employing a splev function. The curve was then $z$-normalized using a zscore function to calculate DTW distances, in which $l\hat{\ }2$-norm is the distance function (only between 80 and 94 °C). Hierarchical clustering based on DTW distances was performed based on a neighbor-joining method to generate a dendrogram.

## DNA profiling by ERIC PCR

ERIC-PCR mixture contained 0.2 μL of extracted *Salmonella* DNA, primer set (Table 1), 2 μL of HOT FIREPol Blend Master Mix Plus 10 mM MgCl₂ (Solis Biodye), and double-distilled water to make 10 μL. Thermocycling was performed in Veriti Thermal Cycler, Applied Biosystems (Thermo Fisher Scientific, Waltham, MA, USA) as follows: 95 °C for 15 min; followed by 30 cycles of 95 °C for 1 min, 54 °C for 2 min and 72 °C for 4 min; and a final step at 72 °C for 7 min. Amplicons were separated by 4% agarose gel-electrophoresis, stained with RedSafe dye (INiRON, Kirkland, WA, USA) and recorded with a Molecular Imager Gel DOC™ XR+ (Bio-Rad, Berkeley, CA, USA) equipped with an Image Lab™ software as JPEG images at 300 dpi resolution.

## Analysis of ERIC PCR amplicon profile and phylogenetic tree

Amplicon patterns generated by ERIC-PCR were analyzed and employing a curve-based algorithm (Pearson correlation) together with an open-source GelJ software (*Heras et al., 2015*) to create a similarity scale, and analysis of constructed phylogenetic tree clusters was performed by an unweighted pair-group using arithmetic averages algorithm (UPGMA).

## CRISPR 2 uniplex and HRM-PCR assays of *Salmonella* isolates

*Salmonella* isolates ($n = 55$) included, in addition, previously characterized *Salmonella* spp. [*S.* 4,[5],12:i:- ($n = 15$), *S.* Typhimurium ($n = 6$) and other *Salmonella* serotypes ($n = 8$)] (*Poonchareon et al., 2019a*; *Poonchareon et al., 2019c*). CRISPR 2 uniplex PCR mixture (10 μL) contained 1 μL of DNA, primer set (IDT, Singapore) (Table 1) and 2 μL of HOT FIREPol Blend Master Mix Plus 10 mM MgCl₂ (Solis Biodye). Thermocycling was carried out as described above but using the following conditions: 95 °C for 12 min; 35 cycles of 94 °C for 60 s, 59 °C for 60 s and 72 °C for 90 s; and a final step at 72 °C for 7 min. Amplicons were analyzed and recorded as described above using 2% agarose gel-electrophoresis. Amplicons of *S.* 4,[5],12:i:- isolates from this study and those previously characterized (nos. 1, 23, 25, 35, 56, 76, 142, 152, 157, and 249) (*Poonchareon et al., 2019a*; *Poonchareon et al., 2019c*) were purified from PCR clean up & gel extraction kit; GeneDirex (Bio-Helix, Keelung City, Taiwan) and sequenced by First BASE Lab (Seri kembangan, Selangor, Malaysia). DNA sequences were aligned and compared with a BioEdit software package (https://bioedit.software.informer.com).

CRISPR 2 HRM-PCR mixture (10 μL) contained 1 μL of DNA, 2 pmol of B1 and 2 pmol of B2 (IDT) (Table 1), and 2 μL of HOT FIREPol EvaGreen: no ROX Mix (Solis Biodye) and thermocycling was performed as described above but under the following conditions: 95 °C for 12 min, followed by 35 cycles of 95 °C for 60 s, 59 °C for 60 s and 72 °C for 90 s. Samples were then heated at 95 °C for 1 min, cooled to 40 °C for 1 min and HRM profiles generated and analyzed as described above but using HRM profile of *S.* Agona as baseline. Clustering of *S.* 4,[5],12:i:- isolates was achieved with a pre-melt normalization range of 87.7–88.2 °C and a post-melt normalization range of 91.5–91.7 °C, with *S.* 4,[5],12:i:- isolate 76 as a reference isolate.

## Characterization of CRISPR 2 amplicon sequences

CRISPR 2 DNA sequences in fasta format were uploaded to https://crisprcas.i2bc.paris-saclay.fr/CrisprCasFinder/Index with default setting. Information of CRISPR 2 region (molecular signatures of CRISPR 2 locus), e.g., DR consensus sequence, DR length and number of spacers, were provided directly in the form of DNA sequences or numeral data.

## Identification of *bla* alleles

Reaction mixture (10 μL) for multiplex PCR of *bla* alleles of interest contained 1 μL of DNA, *bla* primer sets (IDT) (Table 1) and 2 μL of HOT FIREPol Blend Master Mix Plus 10 mM MgCl$_2$ (Solis Biodye) and thermocycling was conducted as described above but using the following conditions: 95 °C for 12 min; 30 cycles of 95 °C for 40 s, 60 °C for 40 s and 72 °C for 60 s; and a final step at 72 °C for 7 min. Amplicons were analyzed (using 1.5% agarose gel) and recorded as described above.

# RESULTS

## *Salmonella* serotypes of clinical isolates from stools of diarrheal patients, Phayao Ram Hospital, Phayao province

Multiplex HRM-PCR of *Salmonella flj*B, *gyr*B and *ycf* Q amplicons from 26 patients, stool sample isolates generated 11 distinct HRM profiles (HRM_1-11) with HRM_1 predominant ($n = 10$, indicative of *S.* 4,[5],12:i:-), followed by HRM_6 ($n = 4$, indicative of *S.* Enteritidis) and HRM_8 ($n = 3$, indicative of *S.* Stanley) (Fig. 1). HRM_5 profile ($n = 2$) was similar to those previously identified for *Salmonella* serotypes *S.* Agona, *S.* Corvallis, *S.* Derby, and *S.* Kedougou (*Poonchareon et al., 2019b*). The other seven HRM profiles were not characterized and considered as indicative of rare or unknown *Salmonella* serotypes.

ERIC PCR was applied to characterize genetic relatedness of *S.* Enteritidis (HRM_6), HRM_2, _3, _4, _5, _7, _9, _10, and _11 *Salmonella* isolates. Phylogenetic tree constructed from ERIC PCR amplicon sizes revealed these *Salmonella* isolates clustered into three distinct clades (Fig. 2). *Salmonella* isolates 378 and 454 (HRM_5) were in clade 1 and assigned *S.* Weltevreden. All *S.* Enteritidis isolates (HRM_6) were grouped in clade 2, and *Salmonella* isolates with serotypes unidentified from multiplex HRM-PCR profiles were grouped together in clade 3 and still remained untyped except for *Salmonella* isolate 412 (HRM_4) that was assigned *S.* 4,[5],12:i:-.

## Phylogenic tree based on multiplex HRM-PCR of clinical *Salmonella* isolates and association with other characteristics

A DTW algorithm was applied to transform HRM-PCR profiles into numerical data to allow construction of a phylogenetic tree (*Lu et al., 2017*), resulting in grouping of the 26 clinical isolates into six clades (Fig. 3). *S.* 4,[5],12:i:-, *S.* Enteritidis and *S.* Stanley was clustered in clade 2, 4 and 6, respectively while HRM types of unidentified serotypes were mainly distributed in clade 3 and 5 except HRM_5 (*S.* Weltevreden) and isolate 412 (HRM_4, *S.* 4,[5],12:i:-) was in clade 1 and 3 respectively. It is worth noting that *S.* 4,[5],12:i:- isolates were clustered in clades 2 ($n = 8$) and 3 ($n = 1$), but three were

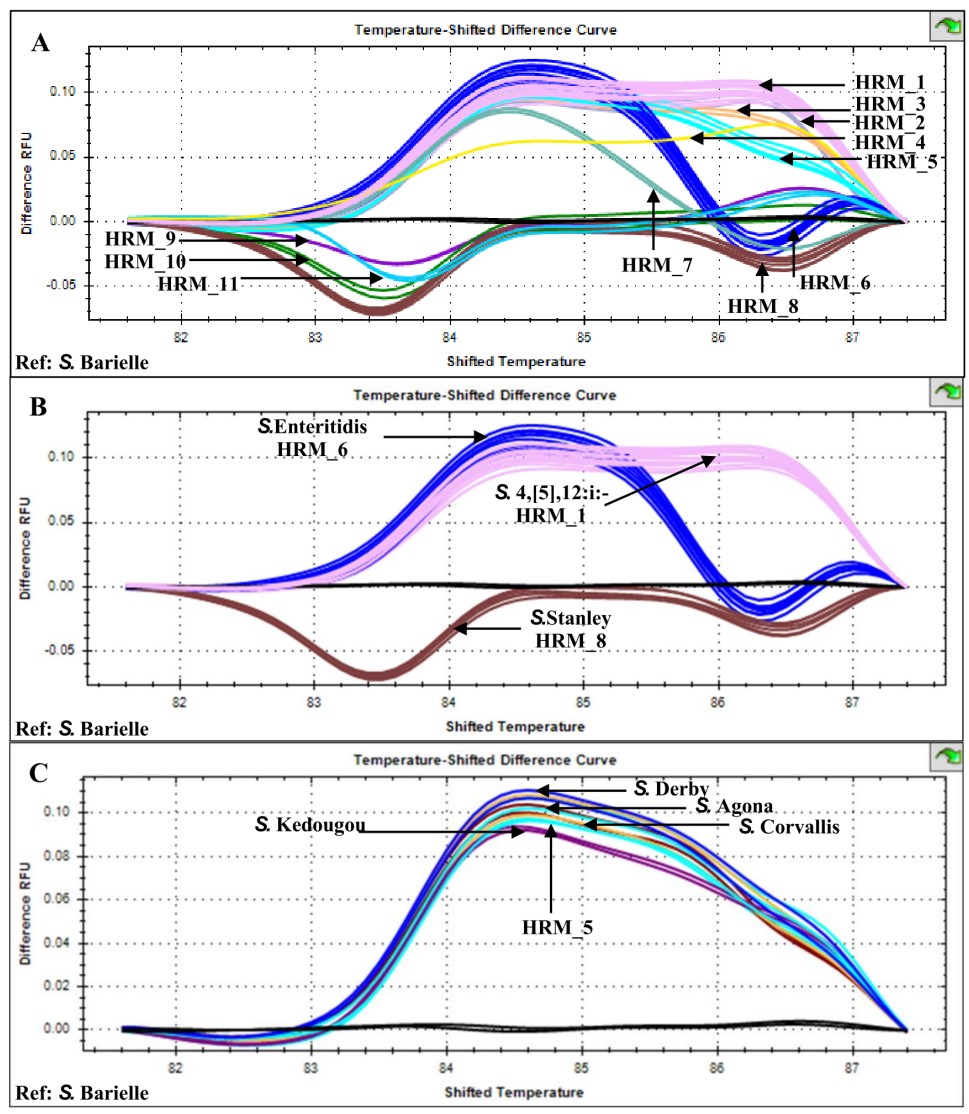

**Figure 1** High resolution temperature-shifted difference melting profiles of multiplex PCR amplicons from clinical *Salmonella* isolates of diarrheal patients' stools (*n* = 26), Phayao Ram Hospital, Phayao province, Thailand (2016–2017). Multiplex PCR of *fljB*, *gyr*B and *ycfQ* was carried out employing primer pairs listed in Table 1. Temperature-shifted difference profiles relative to *S.* Bareilly were produced using a Precision Melt Analysis software V 1.2 (BIO-RAD, Hercules, CA, USA). (A) HRM profiles of *Salmonella* isolates (assigned HRM 1-11). (B) HRM profiles of the three most prevalent HRM profiles and their assigned *Salmonella* serotypes. (C) HRM_5 profile compared to known *Salmonella* serotypes (*Poonchareon et al., 2019b*). RFU, relative fluorescence unit.

unassigned. Ampicillin resistance was predominantly associated with *bla*TEM, and two ESBL-producing *Salmonella* isolates (based on both double-dish and the rapid ESBL assay) harbored *bla*CTX group 1. One ESBL-producing *S.* 4,[5],12:i:-, isolate 412 (assigned by ERIC-PCR, Fig. 2) demonstrated a multidrug-resistant phenotype, including CIP resistance.

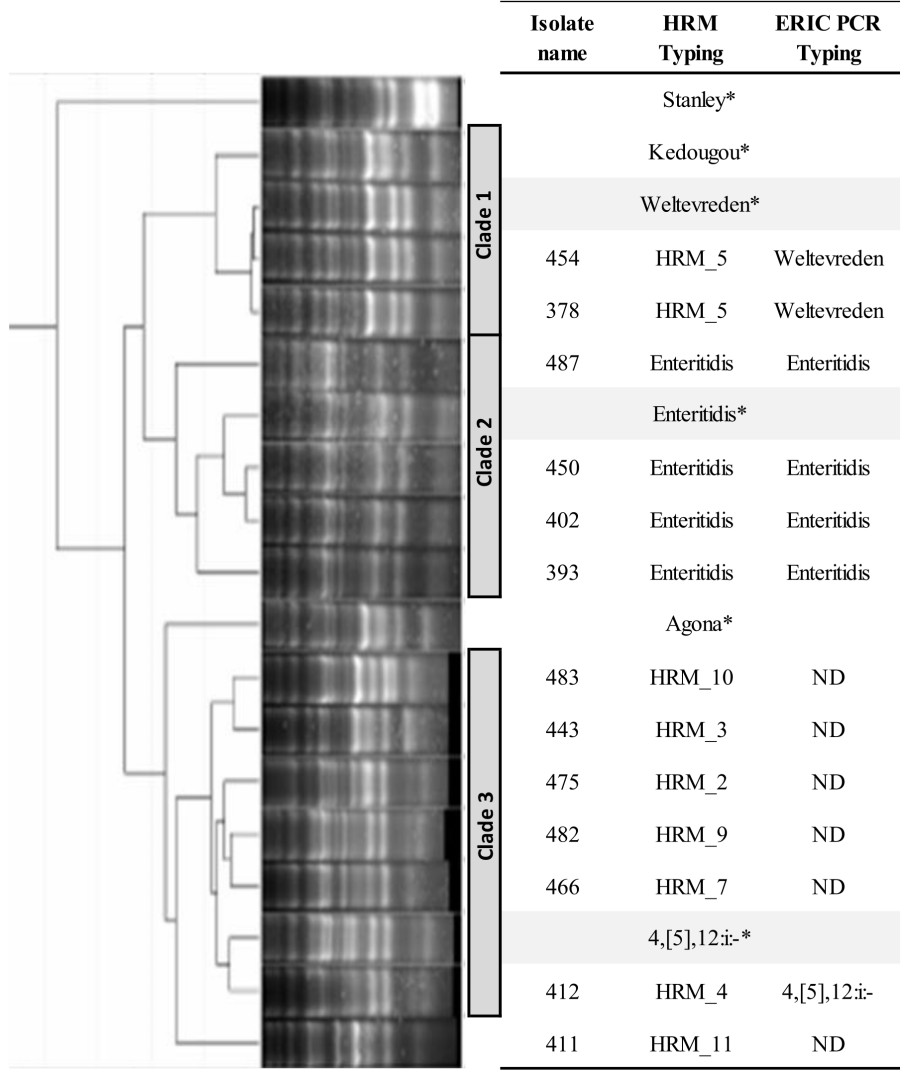

| Isolate name | HRM Typing | ERIC PCR Typing |
|---|---|---|
| Stanley* | | |
| Kedougou* | | |
| Weltevreden* | | |
| 454 | HRM_5 | Weltevreden |
| 378 | HRM_5 | Weltevreden |
| 487 | Enteritidis | Enteritidis |
| Enteritidis* | | |
| 450 | Enteritidis | Enteritidis |
| 402 | Enteritidis | Enteritidis |
| 393 | Enteritidis | Enteritidis |
| Agona* | | |
| 483 | HRM_10 | ND |
| 443 | HRM_3 | ND |
| 475 | HRM_2 | ND |
| 482 | HRM_9 | ND |
| 466 | HRM_7 | ND |
| 4,[5],12:i:-* | | |
| 412 | HRM_4 | 4,[5],12:i:- |
| 411 | HRM_11 | ND |

**Figure 2** **Phylogeny tree of repetitive element-based (ERIC)-PCR amplicon size profiles of *Salmonella* isolates from diarrheal patients' stools, Phayao Ram Hospital, Phayao province, Thailand (2016–2017).** ERIC-PCR was performed using primers listeded in Table 1. The phylogeny tree was constructed using an unweighted pair-group using arithmetic averages algorithm. Clustering into clades employed a curve-based algorithm (Pearson correlation) together with an open-source GelJ software (*Heras et al., 2015*). *Known *Salmonella* serotypes. HRM, multiplex high resolution melting PCR; ND, no designation of serotype.

## CRISPR 2 uniplex and HRM-PCR analyses

In order to simplify *Salmonella* typing based on multiplex PCR of *flj*B, *gyr*B and *ycf* Q, uniplex PCR of CRISPR 2 locus was developed. The 26 clinical *Salmonella* isolates each generated a single amplicon, size of which was indicative of serotype when compared with known *Salmonella* serotypes, namely, 830 bp of *S.* Enteritidis and 1,700 bp of *S.* 4,[5],12:i:- (Fig. 4). *Salmonella* isolate 412, shown by ERIC-PCR to be *S.* 4,[5],12:i:-, generated an amplicon of 1,700 bp. However, samples 378 and 454 (*S.* Weltevreden) and 312, 413 and

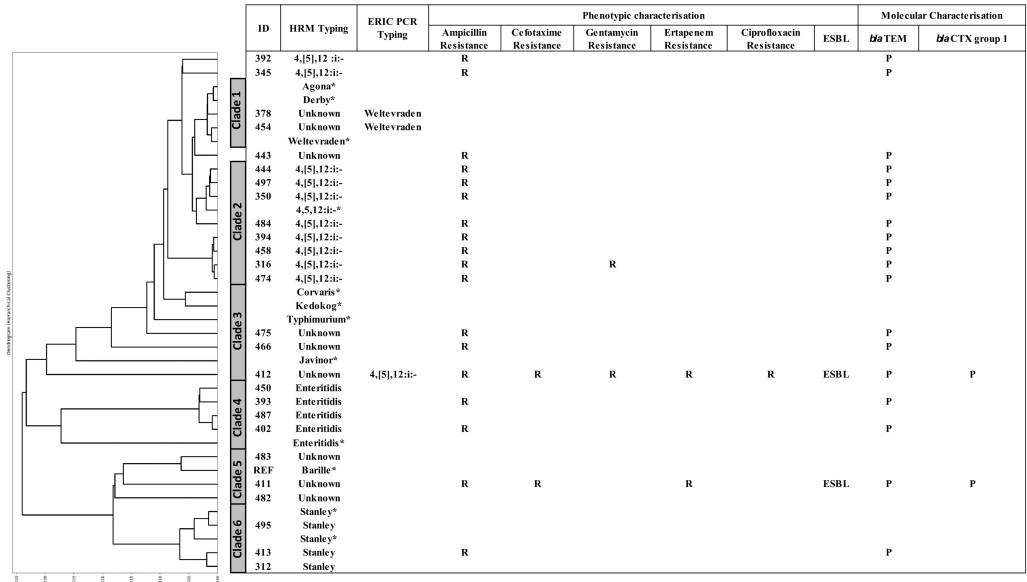

**Figure 3** **Phylogeny tree based on multiplex high resolution melting (HRM) PCR profiles of *Salmonella* isolates (*n* = 26) from diarrheal patients' stools, Phayao Ram Hospital, Phayao province, Thailand (2016–2017).** Phylogeny tree was constructed using a dynamic time warping algorithm (Python platform). Properties of the clinical *Salmonella* isolates are also shown. *Salmonella* isolates of known serotypes are marked with an asterisk. ESBL, extended spectrum β-lactamase producer; ERIC PCR Typing, repetitive element-based PCR amplicon size profile; HRM Typing, multiplex HRM PCR profiling; ID, sample number; P, positive; R, resistant.

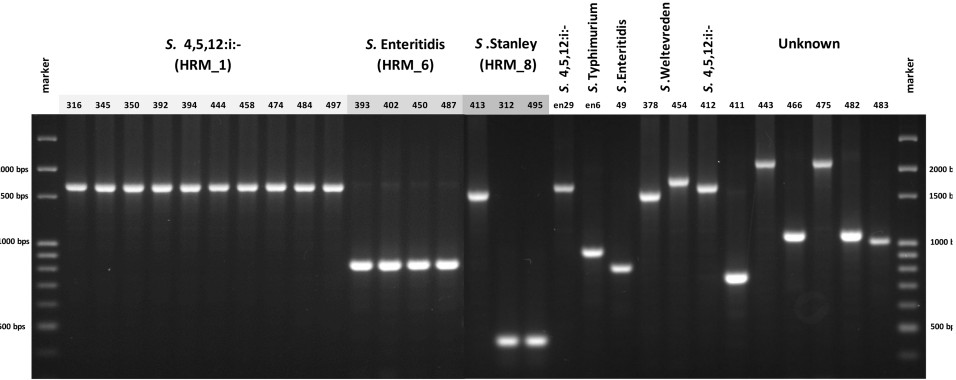

**Figure 4** **Gel electrophoresis of CRISPR 2 uniplex PCR amplicons of *Salmonella* isolates (*n* = 26) from diarrheal patients' stools, Phayao Ram Hospital, Phayao province, Thailand (2016–2017).** Uniplex PCR was performed using primers listed in Table 1. Number indicates sample ID. Name/number in box indicates *Salmonella* isolates of known serotypes. marker, DNA size markers.

495 (*S.* Stanley) failed to demonstrate amplicons of consistent sizes. Amplicon sizes of the remaining samples did not correspond to the three *Salmonella* serotype standards. Thus, CRISPR 2 uniplex PCR, although simpler to perform, had limited ability in providing definitive identification of *Salmonella* serotypes based on amplicon sizes.

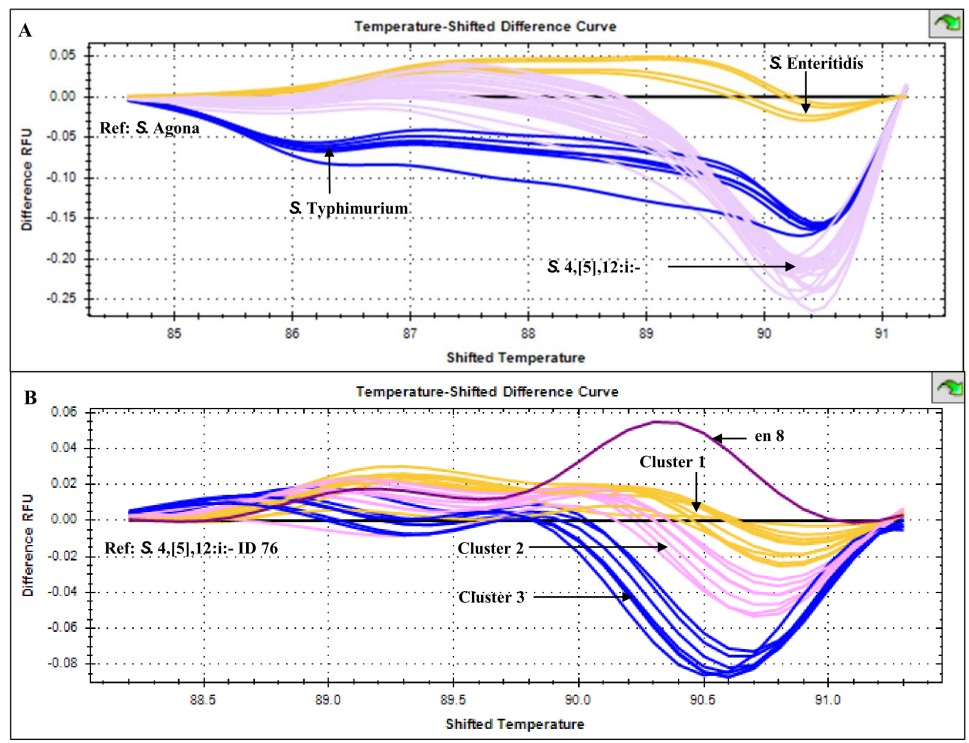

**Figure 5  High resolution melting temperature-shifted difference profiles of CRISPR 2 uniplex amplicons of *Salmonella* isolates from diarrheal patients' stools, Phayao Ram Hospital, Phayao province, Thailand (2016–2017).** Temperature-shifted difference melting profiles relative to (A) S. Agona and (B) *S.* 4,[5],12:i:- isolate 76 were generated using a BIO-RAD CFX96™ Real-Time System (Bio-Rad, Hercules, CA, USA). In (A), S. Typhimurium samples ($n = 5$) were from known stocks; in (B) S. 4,[5],12:i:- samples included clinical samples from this study ($n = 11$), other human samples ($n = 9$) and pork meat ($n = 4$) (see Table 2). en 8, *S.* 4,[5],12:i:- isolate from minced pork; RFU, relative fluorescence unit.

HRM temperature-shifted difference profiles (relative to *S.* 4,[5],12:i:- isolate 76) of CRISPR 2 uniplex amplicons of the 11 *S.* 4,[5],12:i:- isolates from this study and other previously identified samples produced three different profiles (clusters 1-3) (Fig. 5). Cluster 1 contained five known human and four from minced pork samples, cluster 2 four isolates from this study and three known human isolates and cluster 3 seven isolates from this study (including *Salmonella* isolate 345 and 392 that were not grouped in any of the six clades inferred from multiplex HRM PCR profiling (Fig. 3) and *Salmonella* isolate 412) and one known human isolate (Table 2). The three ESBL-producing *Salmonella* clinical isolates in cluster 1 carried a novel variant DR consensus sequence (*Fabre et al., 2012*). Interestingly, ESBL-producing *Salmonella* isolate 412 (cluster 3), isolated in 2017, was not grouped with the other three ESBL-producing *S.* 4,[5],12:i:- isolates collected earlier, suggesting the possibility of a different evolution trajectory of isolate 412.

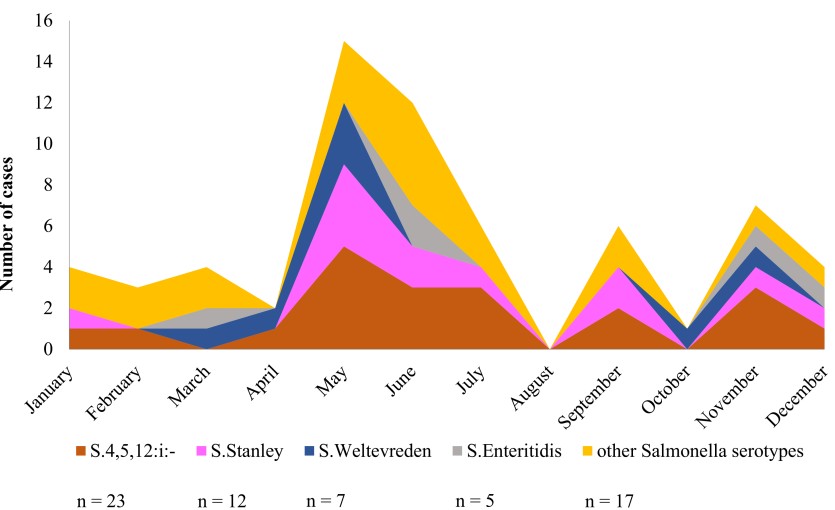

**Figure 6** **Seasonal variation of *Salmonella* isolates from diarrheal patients (*n* = 64) admitted to Phayao Ram Hospital, Phayao province, Thailand (April 2015–July 2017).** *Y*-axis indicates cumulative number of cases in a given month. Serotype was determined using multiplex high resolution melting PCR profiling.

## Seasonal prevalence of *Salmonella* serotypes collected from stools of patients, Phayao Ram Hospital

Over a period of one calendar year, infection at Phayao Ram Hospital of *Salmonella* serotypes identified using multiplex HRM-PCR profiling peaked during the summer period (April to August) (Fig. 6), with, as expected, *S.* 4,[5],12:i:- being predominant, but all serotypes were represented. Although minor peaks were discernable, they are not of significance owing to the limited number of *Salmonella* isolates examined.

## DISCUSSION

In Thailand, *Salmonella* spp. is the leading cause of food poisoning with approximately 359-389/100000 of the population affected (*National Disease Surveiilance, 2017*). In Phayao province, the estimated number of cases of food poisoning increased from 143 to 288/100,000 between 2015 to 2016, but with no reported mortality (*National Antimicrobial Resistance Surveillance Center, 2017*). *Salmonella* spp. isolated from stool of patients, especially from infants, are consistent among studies irrespective of geographical locations (*Pulsrikarn et al., 2013*; *Parry & Threlfall, 2008*). There were 105 cases admitted to Phayao Hospital from *Salmonella* food poisoning from January to December 2014 (*Phayao Hospital, 2015*).

Multiplex HRM-PCR assay of *Salmonella* serotypes, although not able to type all eight serotypes responsible for food poisoning, is adequate in identifying the majority of pertinent clinical *Salmonella* serotypes in particular *S.* 4,[5],12:i:- and *S.* Enteritidis that are frequently identified as the serotypes responsible for human and veterinary infections worldwide (*Frasson et al., 2016*). The method is less time consuming, with a turn-around time of approximately 8–12 h, compared to 3–5 days using standard conventional technique

**Table 2** Properties of *Salmonella* 4,5,12:i- isolates present in clusters based on CRISPR 2 uniplex high resolution melting PCR profiles.

| Cluster | Serotype | ID | Source | Year of isolation | ESBL | Amplicon size (bp) | CRISPR length (bp) | Spacer count | DR concensus sequence | DR Length (bp) | Group (% Identity) |
|---|---|---|---|---|---|---|---|---|---|---|---|
| Cluster 1 | 4,[5],12:i:- | 1* | Human | 2015 | | 1,959 | 579 | 21 | CGGTTTATCCCCGCTGGCGCGGGGAACAC[1] | 29 | 3, 2(99.45), 1(50.69) |
| | 4,[5],12:i:- | 25* | Human | 2015 | | 1,959 | 1,493 | 24 | CGGTTTATCCCCGCTGGCGCGGGGAACAC[1] | 29 | 1 (100) |
| | 4,[5],12:i:- | 152* | Human | 2015 | + | 1,959 | 1,493 | 24 | GTGTTCCCCGCGCCAGCGGGGATAAACCG[2] | 29 | 2 (100) |
| | 4,[5],12:i:- | 157* | Human | 2015 | + | 1,959 | 1,493 | 24 | GTGTTCCCCGCGCCAGCGGGGATAAACCG[2] | 29 | 2 (100) |
| | 4,[5],12:i:- | 249* | Human | 2016 | + | 1,959 | 1,493 | 24 | GTGTTCCCCGCGCCAGCGGGGATAAACCG[2] | 29 | 2 (100) |
| | 4,[5],12:i:- | en11* | Minced pork | 2017 | | | | | | | |
| | 4,[5],12:i:- | en21* | Minced pork | 2017 | | | | | | | |
| | 4,[5],12:i:- | en20* | Minced pork | 2017 | | | | | | | |
| | 4,[5],12:i:- | en29* | Minced pork | 2017 | | | | | | | |
| Cluster 2 | 4,[5],12:i:- | 23* | Human | 2015 | | 1,959 | 1,493 | 21 | CGGTTTATCCCCGCTGGCGCGGGGAACAC[1] | 29 | 1 (100) |
| | 4,[5],12:i:- | 56* | Human | 2015 | | 1,959 | 1,493 | 24 | CGGTTTATCCCCGCTGGCGCGGGGAACAC[1] | 29 | 1 (100) |
| | 4,[5],12:i:- | 142* | Human | 2015 | | 1,959 | 1,493 | 21 | CGGTTTATCCCCGCTGGCGCGGGGAACAC[1] | 29 | 1 (100) |
| | 4,[5],12:i:- | 444 | Human | 2017 | | | | | | | |
| | 4,[5],12:i:- | 474 | Human | 2017 | | | | | | | |
| | 4,[5],12:i:- | 484 | Human | 2017 | | | | | | | |
| | 4,[5],12:i:- | 497 | Human | 2017 | | | | | | | |
| Cluster 3 | 4,[5],12:i:- | 35* | Human | 2015 | | 1,959 | 1,493 | 24 | GTGTTCCCCGCGCCAGCGGGGATAAACCG[2] | 29 | 2 (100) |
| | 4,[5],12:i:- | 316 | Human | 2016 | | | | | | | |
| | 4,[5],12:i:- | 345 | Human | 2016 | | | | | | | |
| | 4,[5],12:i:- | 350 | Human | 2016 | | | | | | | |
| | 4,[5],12:i:- | 392 | Human | 2016 | | | | | | | |
| | 4,[5],12:i:- | 394 | Human | 2016 | | | | | | | |
| | 4,[5],12:i:- | 412 | Human | 2017 | + | | | | | | |
| | 4,[5],12:i:- | 458 | Human | 2017 | | | | | | | |
| | 4,[5],12:i:- | en8* | Minced pork | 2017 | + | | | | | | |
| Ref | 4,[5],12:i:- | 76* | Human | 2015 | + | 1,959 | 1,493 | 24 | CGGTTTATCCCCGCTGGCGCGGGGAACAC[1] | 29 | 1(100) |

**Notes.**

*, S. 4,[5],12:i:- isolates from previous study (method section); DR, direct repeat; ESBL, extended spectrum beta lactamase; ref, reference strain; Spacer count, number of spacers in CRISPR 2 locus; similar DNA sequence of DR concensus labelled with the same superscript number.
and is less complicated to perform. A number of different primers have been employed in multiplex PCR for geno and sero *Salmonella* spp. (*O'Regan et al., 2008*; *Kim et al., 2006*), but the numbers have been kept to a minimum owing to the vast variety of HRM profiles that could be generated (*Masek et al., 2014*).

*S.* 4,[5],12:i:- is considered a major virulent *Salmonella* serotype that has successfully dispersed throughout many geographical areas and harbored various groups of antibiotic resistance genes (*Guerra et al., 2000*). In the present study, most of the *Salmonella* spp. collected from diarrheal patients were resistant to only ampicillin, although the identification of an ESBL- producing *S.* 4,[5],12:i:- suggests a possible contamination of the chain of food production originating from swine and chicken farms in this region (*Angkititrakul et al., 2005*). ESBL-producing and MDR *S.* 4,[5],12:i:- harboring virulent *bla*CTX group 1 is frequently found in other regions of Thailand (*Amavisit, Boonyawiwat & Bangtrakulnonth, 2005*) and in Southeast Asian countries (*Tamang et al., 2011*). Several rapid ESBL tests are available with demonstrated superiority in reducing operational time, simplicity in procedure and high accuracy (*Nordmann, Dortet & Poirel, 2012*). A convenient and rapid ESBL test based on observed color change of direct samples was employed in our study without any discrepancies previously reported (*Srisrattakarn et al., 2016*). Because the commonly prescribed drugs for gastroenteritis treatment are the quinolone and $\beta$-lactam-related antibiotics, emergence of MDR *Salmonella* strains resistant to both ciprofloxacin and cefotaxime has compromised use of the drugs in cases with complications, such as bacteremia with underlying diseases or in children (*Guarino, Bruzzese & Giannattasio, 2018*). Association of resistance to a particular $\beta$-lactam antibiotic with a *Salmonella* serotype highlights the usefulness of rapid *Salmonella* typing and the emergence of *S.* 4,[5],12:i:- resistant in both ciprofloxacin and cefotaxime indicates the need for implementation of surveillance in local hospitals. Although the peak season for *Salmonella* infection in the study region was in the summer season, consistent predominance of virulent *S.* 4,[5],12:i:- among the isolates throughout the year is be of concern.

CRISPR polymorphism has been effectively applied in typing and subtyping several clinically important bacteria such as *Mycobacterium tuberculosis* and *Salmonella* spp. (*Kamerbeek et al., 1997*; *Liu et al., 2011*). A significantly improved CRISPR-based method (CRISPOL) for *Salmonella* subtyping was effectively applied to characterize more than 2,000 isolates of *Salmonella* Typhimurium and variant *S.* 4,[5],12:i:- (*Fabre et al., 2012*). However, a simple and rapid conventional PCR assay was also developed to characterize certain specific serotypes (*Xiong et al., 2016*). Our small cohort study indicates a preliminary protocol of CRISPR 2 uniplex PCR allowed simultaneous detection and differentiation among three clinically important *Salmonella* serotypes, namely *S.* 4,[5],12:i:-, *S.* Enteritidis and *S.* Typhimurium but this needs confirmation using a larger collection of *Salmonella* isolates. This CRISPR 2 uniplex PCR method should be less complicated to adapt in quantitative analysis of *Salmonella* contamination in food products compared to quantitative multiplex PCR (*Maurischat et al., 2015*). In addition, application of CRISPR HRM-PCR profiling to address CRISPR 1 and 2 sequence polymorphisms has allowed differentiation of eleven *Salmonella* serotypes (*Bratčikov & Mauricas, 2009*). CRISPR

2 HRM-PCR profiling in our hands indicated *S.* 4,[5],12:i:- isolates could be readily differentiated, possibly reflecting differences in evolutionary trajectories of the *S.* 4,[5],12:i:- isolates from this region of the country.

The limited number of *S.* 4,[5],12:i:- isolates available in this particular area necessitates further studies of a larger cohort of *S.* 4,[5],12:i:- isolates not only from the study area but from various epidemiology backgrounds to enable correlation of important virulent features with epidemiological settings. Interestingly, application of CRISPR 1 and 2 PCR methods for *Salmonella* identification allowed detection of multiple *Salmonella* serovars in a single sample (*Thompson et al., 2018*). This application of CRISPR 2 PCR for identification of multiple *Salmonella* serotypes in a specimen will be further explored.

## CONCLUSIONS

The study demonstrates the ability of applying multiplex HRM-PCR profiling and a novel protocol of CRISPR 2 uniplex PCR together with HRM profiling to rapidly and conveniently type clinically important *Salmonella* isolates. In conjunction with a rapid ESBL test, antibiogram profile and detection of $\beta$-lactamase genes should provide a powerful strategy for predicting antibiotic resistance phenotypes associated with particular *Salmonella* serotypes. Moreover, CRISPR 2 HRM-PCR profiling provides a novel tool for classification of *S.* 4,[5],12:i:- according to evolutionary trajectories stemming from various epidemiological settings.

## ACKNOWLEDGEMENTS

The authors appreciated research cooperation from Veterinary Research and Development Center (Upper Northern Region). Finally, we were indebted to Prof. Emeritus Prapon Wilairat, Mahidol University for providing invaluable suggestions and encouragement and for proof-reading the final manuscript.

### Funding
The author received no funding for this work.

### Competing Interests
The authors declare there are no competing interests.

### Author Contributions
- Nuttachat Wisittipanit conceived and designed the experiments, analyzed the data, prepared figures and/or tables, dTW analysis, and approved the final draft.
- Chaiwat Pulsrikarn performed the experiments, prepared figures and/or tables, conventional Serotyping of Standard *Salmonella* spp., and approved the final draft.
- Sudarat Srisong performed the experiments, prepared figures and/or tables, CRISPR 2 PCR for *Salmonella* typing, and approved the final draft.

- Rungthiwa Srimora performed the experiments, prepared figures and/or tables, hRM PCR for Salmonlla typing, and approved the final draft.
- Nattinee Kittiwan performed the experiments, prepared figures and/or tables, confirm *Salmonella* Serotyping, and approved the final draft.
- Kritchai Poonchareon conceived and designed the experiments, performed the experiments, analyzed the data, prepared figures and/or tables, authored or reviewed drafts of the paper, conceptual design of all experiments, and approved the final draft.

## Human Ethics

The following information was supplied relating to ethical approvals (i.e., approving body and any reference numbers):

The study was conducted with the ethical approval: NO 57 02 04 0020, from the ethical committees of the University of Phayao.

## Data Availability

Raw data is available in the Supplemental Files.

## Supplemental Information

Supplemental information for this article can be found online at http://dx.doi.org/10.7717/peerj.9113#supplemental-information.

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
