# Peer review of "CRISPR 2 PCR and high resolution melting profiling for identification and characterization of clinically-relevant Salmonella enterica subsp. enterica"

_PeerJ, doi:10.7717/peerj.9113_

## Round 0.1 · original submission · Major Revisions

Please see the reviewer's comments and I appreciate you considering them to improvise the manuscript. I do feel the manuscript requires more details on CRISPR methodology used. and please do highlight the novelty of the study and the limitations of the study as well. I strongly recommend getting your manuscript read and reviewed by a Science English expert before resubmission.

Reviewer 1 ·

Basic reporting

No comment.

Experimental design

No comment.

Validity of the findings

No comment.

Additional comments

This study has been well done. However, Major revisions are needed. First, there are many errors of program and syntax in the manuscript which should be entirely revised. Second, the originality of the work is very limited. The authors have performed quite similar studies on the serotyping of Salmonella with similar methods in recently published articles. Third, it is not easy to understand the inconsistence of the number of Salmonella isolates (Line 27:26 isoaltes, Line 36: 68 isolates,Figure 7: 64 patients).

Annotated reviews are not available for download in order to protect the identity of reviewers who chose to remain anonymous.

Reviewer 2 ·

Basic reporting

Wisitipanit and colleagues describe in their paper a technic to type and characterize Salmonella enterica subsp. enterica Enteritidis, Typhimurium and its monophasic variant. The text is well-written and the introduction brings the sufficient basis for the reader to understand the context.

Experimental design

The experiments are very well described and detailed.

Validity of the findings

no comment

Additional comments

The text should be amended to have the species and the gene name in italics.

Reviewer 3 ·

Basic reporting

The English language in the manuscript should be improved to clearly understand what the authors wanted to say. Grammatically it should also be very sound.

Experimental design

I have no issues with the authors experimental design for this study.

Validity of the findings

In materials & methods, the authors mentioned that they used 28 stool samples (line 107) but in the results section they only mentioned about 26 isolates (lines 252,294, 310). They also mentioned 16 isolates dominated in 3 HRM profiles but the numbers they provided did not prove that (line 253-254). It is suggested that they should check with all the numbers and percentages they provided in first paragraph of the results section.
In line 264 while explaining figure 2, they wrote isolates 458 and 378 but in actual figure it says 454.
Figure 4 does not have full legend.
In figure 5 legend, they mentioned about 25 clinical isolates but in the results sections they wrote 26 isolates. They also did not show A and B in their original figure.
Figure 6 was not labeled as A, B and C.

Additional comments

In the discussions, the authors need to provide more explanations about why their methods are better than the existing methods. They also need to write in more detailed way the advantages and limitations of their assay. They also should mention what can be done to improve it further. In the discussions, I would suggest them to describe the scope of their assays.

---

## Round 0.2 · accepted · Accept

Thank you for your resubmission and modifying the manuscript based on the reviewers comments. I am happy to recommend the revised version for publication in PeerJ. Best wishes

Reviewer 1 ·

Basic reporting

no comment

Experimental design

no comment

Validity of the findings

no comment

Additional comments

not comments

Reviewer 3 ·

Basic reporting

The language is very clear and professional this time.

Experimental design

No comment.

Validity of the findings

No comment.

Additional comments

The authors have put so much effort to make the manuscript a better one than the previous version. The manuscript is much better understood with all corrections that the reviewers suggested. They also added legends and modified some figures based on reviewers' comments. This version of manuscript is a good read now.